# Consecutive and Effective Facial Masking Using Image-Based Bone Sensing for Remote Medicine Education

**Sinan Chen** [1,2,*] , **Masahide Nakamura** [1,3,*] **and Kenji Sekiguchi** [4]

1. Center of Mathematical and Data Sciences, Kobe University, 1-1 Rokkodai-cho, Nada, Kobe 657-8501, Japan
2. Japan Society for the Promotion of Science, 5-3-1 Kojimachi, Chiyoda-ku, Tokyo 102-0083, Japan
3. RIKEN Center for Advanced Intelligence Project, 1-4-1 Nihonbashi, Chuo-ku, Tokyo 103-0027, Japan
4. Graduate School of Medicine, Kobe University, 1-1 Rokkodai-cho, Nada, Kobe 657-8501, Japan
* Correspondence: chensinan@ws.cs.kobe-u.ac.jp (S.C.); masa-n@cs.kobe-u.ac.jp (M.N.);
   Tel.: +81-78-803-6295 (S.C.)

**Abstract:** Unlike masking human faces from images, facial masking in real-time, frame by frame from a video stream, presents technical challenges related to various factors such as camera-to-human distance, head direction, and mosaic schemes. In many existing studies, expensive equipment and huge computational resources are strongly required, and it is not easy to effectively realize real-time facial masking with a simpler approach. This study aims to develop a secure streaming system to support remote medicine education and to quantitatively evaluate consecutive and effective facial masking using image-based bone sensing. Our key idea is to use the facial feature of bone sensing instead of general face recognition techniques to perform facial masking from the video stream. We use a general-purpose computer and a USB fixed-point camera to implement the eye line mosaic and face mosaic. We quantitatively evaluate the results of facial masking at different distances and human head orientations using bone sensing technology and a depth camera. we compare the results of a similar approach for face recognition with those of bone sensing. As the main results, consecutive face masking using bone sensing is unaffected by distance and head orientation, and the variation width of the mosaic area is stable within around 30% of the target area. However, about three-fourths of the results using conventional face recognition were unable to mask their faces consecutively.

**Keywords:** facial masking; image; bone sensing; video stream; real-time processing; eye line mosaic; face mosaic; face recognition

## 1. Introduction

In recent years, the spread of COVID-19 has had a severe and profound impact on various areas, including the daily lives of individuals, education at schools, and work in the workplace. In the case of medical education in schools during COVID-19 outbreak, it is challenging to coordinate practical training and exercises with a limited number of students [1]. Courses such as clinical practice in outpatient consultation rooms have had to be moved online. However, a new trial of distance learning using software tools such as Zoom [2] is underway. As we implement a new trial of distance learning using software tools, we are mindful of patient privacy when delivering classes online. Security concerns also require that educational video streaming cannot be broadcast over a global network. In addition, for security reasons [3], it is required that the educational video to be distributed cannot be broadcast on a global network. Against this backdrop, research and development of a secure streaming system for medical education in clinical settings are imminent. To research and develop a secure clinical streaming system for medical education, we first consider the protection of facial privacy in video streaming to be a technical challenge. Recently, image recognition and processing techniques utilizing deep learning have been rapidly developed [4]. However, it has yet to achieve highly accurate face detection using limited computational resources and cost. In particular, there are technical limits to the size and orientation of faces that can be detected in real-time face detection from video.

In some cases, recognition accuracy is reduced when a face exceeds a certain distance from the camera or when a face is not in front of the camera, thus making it impossible to protect privacy [5]. We also consider the security protection of communication for video distribution in face protection a technical challenge.

Our interest is to research and develop secure streaming systems in clinical settings for medical education. For this, we consider the following two technical challenges in this research: (1) Face detection in images and videos has a face size detection limit due to the correlation distance between human and camera [6]. (2) The streaming of video with face mosaics also requires communication security protection [7]. The purpose of this study is to develop a secure streaming system to support remote medicine education and to quantitatively evaluate consecutive and effective facial masking using image-based bone sensing. Our key idea is to integrate Web Real-Time Commutation (WebRTC) [8] and image-based bone sensing with the PoseNet model [9]. As an approach, we first use a USB camera and a general-purpose computer to display a video stream of a clinical site in real-time using a Web browser. Next, we extract features of the human face in the acquired video from the recognition results of the bone sensing PoseNet model. Then, the masking process is applied to the extracted facial features, and "eye line mosaic" and "face mosaic" algorithms are designed. Furthermore, the video stream with mosaic depicted in HTML Canvas [10] is delivered to the Web browser using PeerJS technology [11]. In this way, it is expected to facilitate the implementation of distance learning in medical education while delivering the mosaicked video in real-time in the classroom.

The contribution of the proposed method includes the following three parts: (1) Acquire the video stream in real-time and extract only facial features using bone sensing technology. (2) We design an algorithm to mosaic the eyes and face by utilizing the two-dimensional coordinates of the left and right eyes and ears, respectively. (3) We modify the existing PeerJS bidirectional communication system, to enable unidirectional communication from the clinical site to the classroom. It only sends and receives texts using WebSocket technology [12] for bidirectional communication between the clinical site and the classroom. To verify the effectiveness of the proposed method, we implement a secure video streaming system. Specifically, we first install a general-purpose computer and a USB camera. Next, we install the Node.js library [13] and build an HTTP server [14], a PeerJS server, and a WebSocket server. On the client side, we install a bone sensing PoseNet model. In addition, we designed web screen transitions and created three modules: a home page, a video delivery page, and a video viewing page.

To evaluate the effectiveness of the proposed method from various perspectives, we conducted preliminary experiments on face masking using bone sensing and general face recognition under different viewing angles and interpersonal distances for human faces captured from cameras. Our main experimental results show that the results using bone sensing are more stable at middle and far distances, with less drastic changes in the eye mosaic. In contrast, the results using face recognition show that the eye line mosaic and face mosaic are stable at near distances and when the head is facing forward, but otherwise the mosaic is largely unable to produce a mosaic with a more consecutive effect. Through these results, we clearly show that bone sensing is more effective than face recognition for consecutive face masking by combining various factors such as distance and head direction. The remainder of this paper is organized as follows: In Section 2, we discuss related works on image-based face masking. We provide a detained description of the proposed method in Section 3. A preliminary experiment of evaluating face masking effects using bone sensing and face recognition is presented in Section 4, followed by the conclusions in Section 5.

## 2. Related Work

Implementing high-performance secure streaming system with face masking for remote education has been investigated thoroughly in the medical filed [15]. Table 1 shows a comparison of related work for the face masking. We consider the following three challenges in implementing consecutive and effective face masking: (1) It is difficult to achieve

highly accurate face detection with limited computational resources and cost. (2) Technical barriers exist in terms of correlation distance and viewing angle. (3) Large communication overhead to send raw data to the cloud in real time, making it difficult to protect security. Pervasive secure streaming system with face masking is inseparable from consecutive processing, even during the early and late stages. Qiu et al. [16] propose a framework that adds the loss of service quality to the loss function to ensure the generation of de-identified face images with guided quality preservation for the face privacy protection. Kim et al. [17] propose an image modification mechanism that inserts virtual face images that strictly match a given similarity and uses blockchain as a storage area. There is also a case [18] that studies a facial masking scheme for video surveillance systems to protect individual privacy from videos captured by CCTV cameras. Rajput et al. [19] present a cloud-based approach to securely recognize human activities. They secure only four images, including one motion history image and three depth motion maps, which are highly saving the data overhead.

**Table 1.** A comparison of related work for the face masking.

| Research Authors (Year) | Masking Target | Materials and Methods | Intrusiveness | Contributions |
|---|---|---|---|---|
| Kim et al. [18] (2018) | Face | CCTV cameras, video surveillance, face identification | Exist | Found optimal masking values in case study |
| Kim et al. [17] (2020) | Face | Closed-circuit television, CCTV cameras, video surveillance | Exist | Inserted a virtual face used a blockchain as the storage area |
| Rajput et al. [19] (2020) | Face | RGB-D sensors, deep CNN, cloud service | None | Improved the security-recognition accuracy |
| Zhu et al. [15] (2020) | Face, body key points | De-identification, keypoint preservation, face-swapping | Exist | Though video processing to de-identify subjects |
| Our research in this paper (2022) | Face, eye line | General computer, bone sensing, edge computing | Exist | Validated advantages of bone sensing at multi-views |

Unlike these studies, our approach differs from them. We perform face mosaicing in the video stream in real-time, and the ultimate goal is to facilitate medical education by distributing and viewing the video through a edge computing with Web browser. Face detection must first be performed to achieve facial privacy protection in video distribution [15]. On the one hand, we utilize image-based bone sensing techniques to overcome this limitation, which we believe can more smoothly extract features only from face parts. A typical bone sensing technique is the PoseNet model [20]. It is a trained bone sensing model that uses TensorFlow.js, enabling bone sensing in an offline edge environment. Specifically, input any single 2D image and the XY coordinates of 17 body parts and automatically output recognition accuracy values. By developing and driving the PoseNet program, it can be programmed using JavaScript source code in a Web browser or independent Node.js source code. On the other hand, Web Real-Time Communication (WebRTC) [21] technology can be used to realize video distribution using a Web browser. PeerJS technology is a WebRTC communication server that uses JavaScript libraries. PeerJS technology allows webcam video and audio data to be received between browsers (peer-to-peer). On the other hand, WebSocket technology is a WebRTC communication standard for bidirectional communication between a Web browser and a server. We use PeerJS technology for video distribution and WebSocket for bidirectional text message communication.

## 3. Methodology

This section produces a complete description on the proposed method, implementation, and discussion of the related techniques.

### 3.1. Purpose and Approach

Toward improving the challenges described in Section 2, we aim to propose a method to build a secure video streaming system. Our key idea is to integrate WebRTC and image processing technologies. The results using our proposed method will enable online video streaming in real-time with privacy protection for clinical practice in conventional medical education and will achieve a new form of medical education against COVID-19. Figure 1 shows an example of our vision of a secure clinical streaming system for medical education. We envision two related locations for this system, a hospital, and a classroom, with the internal communication and WebRTC technology acting as intermediaries. Specifically, on the hospital side, a USB camera and a general-purpose computer are installed in each outpatient consultation room to acquire real-time video and audio from patients and doctors. Pre-processing is performed to mask human faces in the acquired video in real-time. The pre-processed real-time video and audio are distributed to the corresponding lecture port in the classroom via the internal communication using WebRTC technology. On the other hand, in the classroom, a general-purpose computer is installed in the lecture room, and medical education is conducted while receiving the video streaming with real-time masking of human faces. The system provides medical education while receiving a video feed that masks the human face in real-time. The technical architecture of the proposed method is shown in Figure 2. The core of the method includes the following three parts. (A1) The core of the proposed method is to extract only the bone features of a face from a real-time video stream using bone sensing technology. (A2) Design-related algorithms for facial masking in video streams using bone facial features. (A3) Setup PeerJS technology for unidirectional communication that distributes the video stream from the hospital to the classroom. The following subsections describe each of the core points in detail.

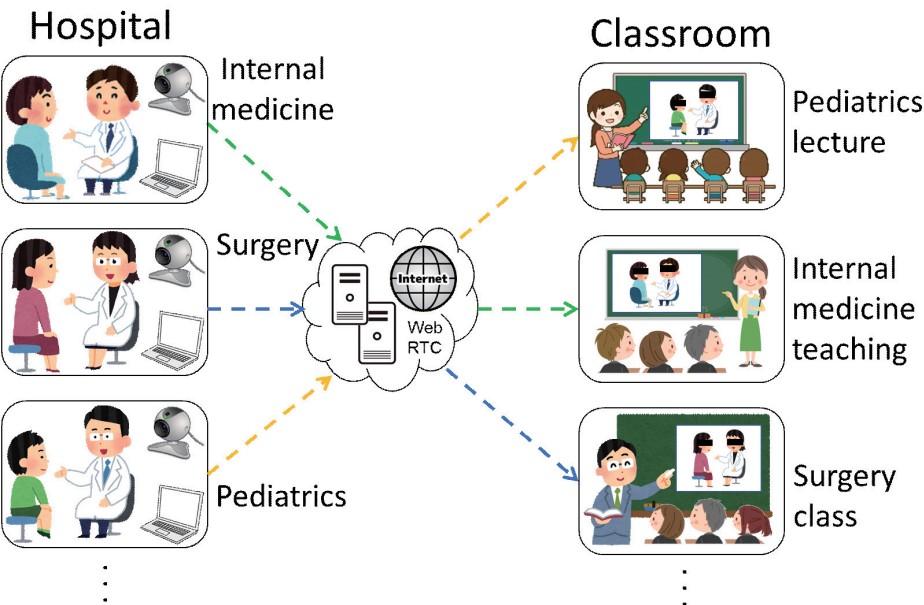

**Figure 1.** A vision of secure streaming systems.

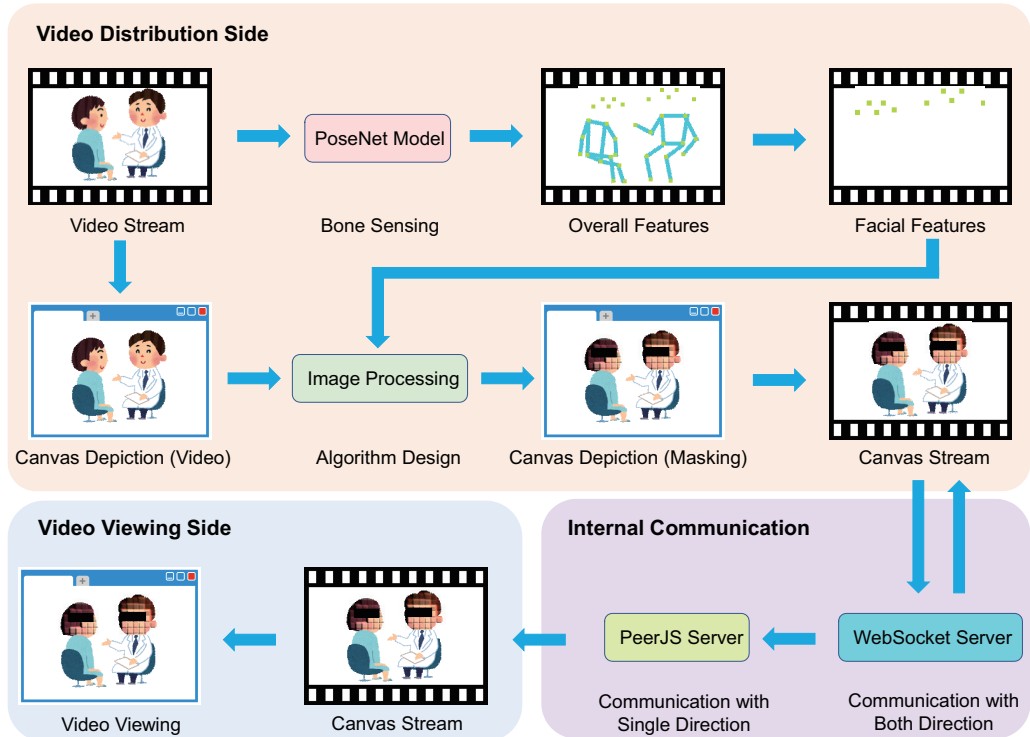

**Figure 2.** Technical architecture in the proposed method.

### 3.2. A1: Feature Extraction in Video Stream

**Step 1: Hardware installation**

In Step 1, a general-purpose computer and a USB fixed-point camera are installed to acquire a video stream of a hospital outpatient examination room. The viewing angle of the USB fixed-point camera can be either flat-angle or wide-angle, but its location should be where it can capture the doctor's and patient's faces.

**Step 2: Acquisition of the video stream**

Step 2 is to display the video stream from the USB fixed-point camera on a Web browser using a Web browser on a general-purpose computer. Because it is necessary to permit the camera connected to the computer to be accessed from the Web browser, the `getUserMedia` method of the Web API's `navigator.mediaDevices` object is used for access.

**Step 3: Introduce bone sensing**

In Step 3, a bone sensing PoseNet model driven by TensorFlow.js is used to obtain the 2D coordinates of the bone keypoints of 17 human body parts in real-time using a Web browser (see Figure 3). The PoseNet model detects a single bone feature or multiple bone features from a person in a video image.

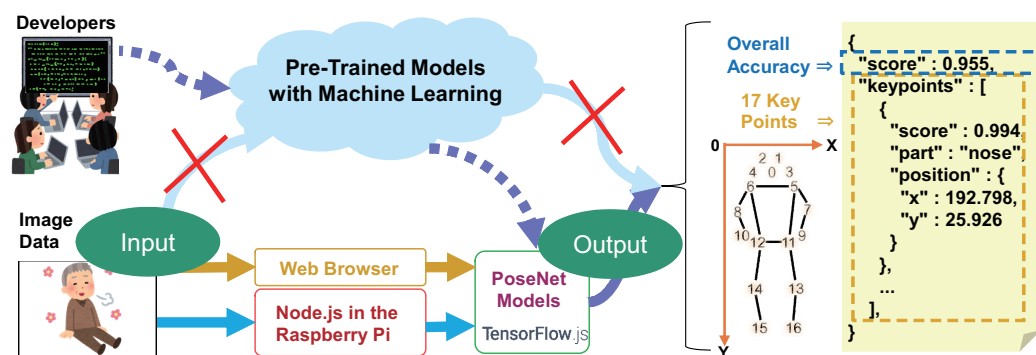

**Figure 3.** Overview of image-based bone sensing using PoseNet models on edge.

*3.3. A2: Facial Masking Algorithm Design*

**Step 4: Extraction of facial bone features**

In Step 4, we extract only five key points of the face (i.e., nose, left eye, right eye, left ear, and right ear) from the 2D coordinates of 17 body parts obtained by the bone-sensing PoseNet model.

**Step 5: Algorithm design for eye mosaic**

In Step 5, we focus on the 2D coordinates of the left eye and the right eye to realize the eye mosaic. Specifically, first, to allow mosaicing up to the limit of the eye's width, we only consider the *x*-axis coordinates for the left eye's two-dimensional coordinates $(x_1, y_1)$ and the right eye's two-dimensional coordinates $(x_2, y_2)$ and extend the distance *d* for the left and right eye, respectively. Extend the distance *d* to each of the left and right eyes. As a result, the 2D coordinates $(x_1 - d, y_1)$ of the new keypoint a and the 2D coordinates $(x_2 + d, y2)$ of the right eye of texttttb are the left and right coordinates of the eye mosaic. Next, we define the thickness *w* of the eye line mosaic. The developer adjusts the values of distance *d* and thickness *w* by checking the video stream of the actual eye mosaic. The concept of a eye line mosaic utilizing eye coordinates is shown in Figure 4a. Finally, using the 2D coordinates and thickness *w* of the keypoints a and b, we execute the program in JavaScript source code that draws black lines in HTML Canvas. On the other hand, in the image captured by the USB fixed-point camera, we consider that the correlation distance between the USB fixed-point camera and the human varies depending on the movement of the human, etc., and we calculate the following. The distance *d* between the left and right extensions of the eye mosaic and the thickness *w* must be fixed values. Hence, we define the length *l* of the eye mosaic (i.e., $l = (x_2 + d) - (x_1 - d) = x_2 - x_1 + 2d$), and the ratio *r* of length *l* and thickness *w* is fixed. The developer adjusts the value of the ratio *r* while checking the video stream of the actual eye line mosaic. Figure 4b shows the concept of eye line mosaic of a face with distance variation. The dynamic length *l* and the fixed ratio *r* are used to calculate the thickness *w* of the eye mosaic and dynamically respond to it.

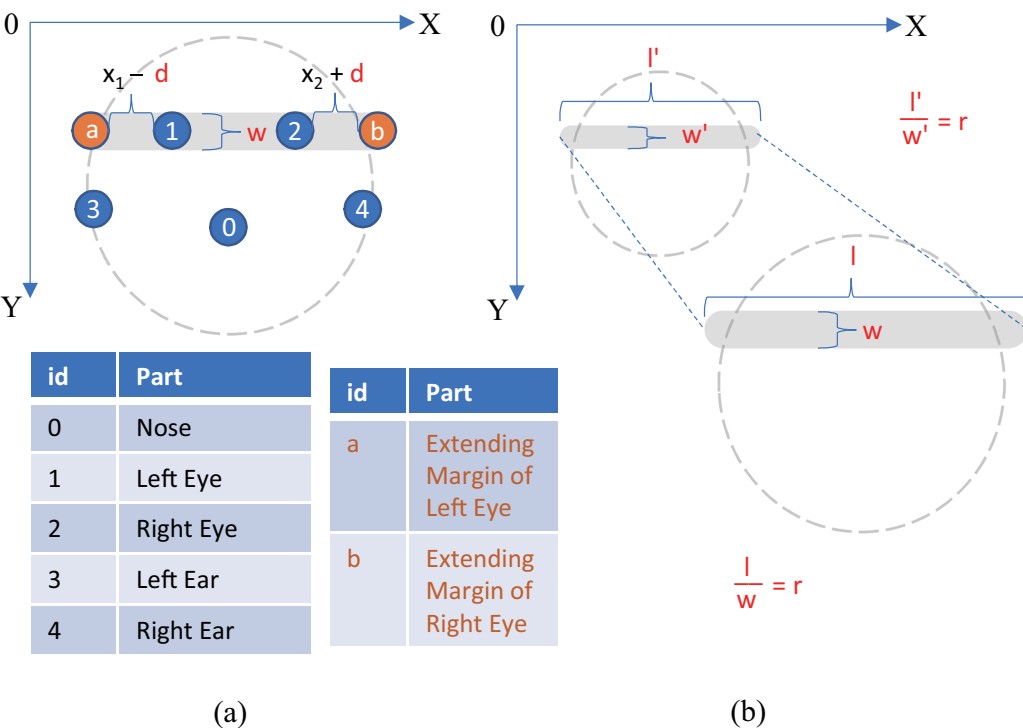

(a)                                            (b)

**Figure 4.** The proposed two concepts for designing algorithm: (**a**) Eye mosaic utilizing eye coordinates. (**b**) Eye line mosaic of face with distance variation.

**Step 6: Algorithm design for face mosaic**

In Step 6, to realize the face mosaic, we focus on the 2D coordinates $(x_3, y_3)$ and $(x_4, y_4)$ of the left and right ears. Specifically, we first describe a circle whose diameter is the correlation distance between the two ears. We define a new keypoint c as the center of the circle, whose two-dimensional coordinates are $((x_4 - x_3)/2, (y_4 - y_3)/2)$. Next, we define the radius *r* of the circle, whose value is $(x_4 - x_3)/2$. The concept of a face mosaic utilizing the ear coordinates is shown in Figure 5a. Using the 2D coordinates and the radius *r* of the circle center c, we crop the original image to the specified circle using the HTML Canvas `clip()` method. We then apply mosaic processing to the clipped circular face image. Figure 5b shows an overview of the face mosaic processing framework. We created a JavaScript source code that loops over the mosaic size in steps, specifically for a circular face image. We first obtain the color information of the corresponding pixel from the pixel information obtained by the method `getImageData`. Next, we draw a square the size of a mosaic using the RGB values of the computed colors. Thus, using the pixel and color information of the face image, a mosaic with square pixels of the defined size is generated.

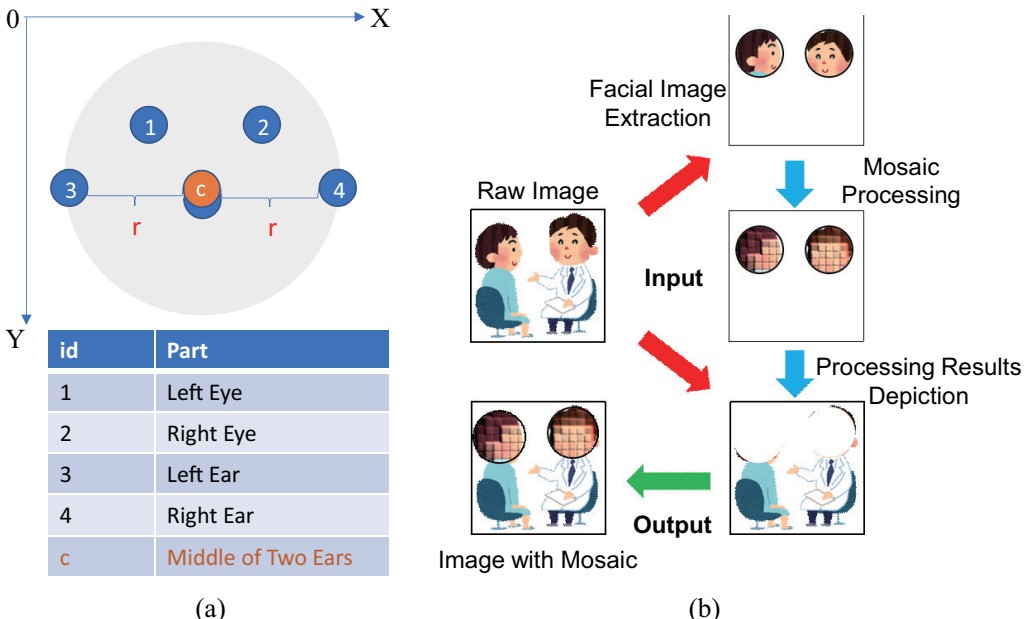

| id | Part |
|----|------|
| 1 | Left Eye |
| 2 | Right Eye |
| 3 | Left Ear |
| 4 | Right Ear |
| c | Middle of Two Ears |

(a)    (b)

**Figure 5.** Overview of the method for face mosaic: (**a**) Concept utilizing ear coordinates. (**b**) Processing framework.

Based on steps 5 and 6, we design special algorithms to achieve facial masking when the head is tilted (see Figure 6). By extracting the slope of the line connected to the 2-dimensional coordinates of the eyes or ears (i.e., $\triangle y/\triangle x = 0$ or not), the proposed system automatically determines whether the head is in a horizontal or non-horizontal position. It will automatically introduce the corresponding algorithm to achieve face masking at different angles. Generally, the human head does not tilt left or right frequently, although head turning is common. Hence, we regard the algorithms in these cases of head tilting as exception handling for some special purposes.

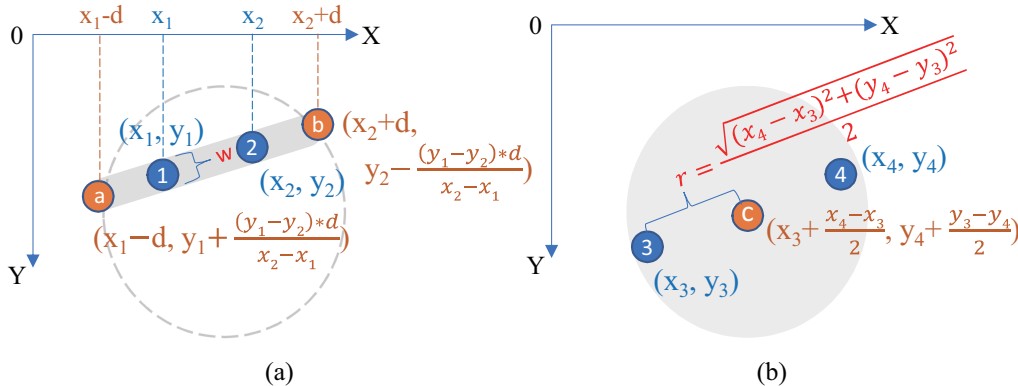

**Figure 6.** Algorithms for retrieving key point coordinates while the head is tilted: (**a**) Eye line mosaic refer to Figure 4. (**b**) Face mosaic refer to Figure 5.

### 3.4. A3: Video Distribution Using WebRTC

**Step 7: Designing Web Screen Transitions**

In Step 7, we divide the web screen of the secure streaming system into three parts: (U1) the home page, (U2) the video streaming page, and (U3) the video viewing page. Figure 7 shows an overview of the Web screen transitions in the proposed method. The proposed system consists of the following five points. (P1) The hospital side starts a new video stream from U1, transitions to U2, and generates the only URL with a random ID. (P2) When U2 is opened, the video streaming starts automatically, and a web link for viewing the U2 video is generated in the distribution list of U1. (P3) The classroom moves to U3 by opening a Web link selected from U1's distribution list. (P4) The video stream to be distributed by U2 is routed to the PeerJS server, which then distributes the video stream to the corresponding U3. (P5) Allows us to send a text stream to either U3 or U2 via a WebSocket server.

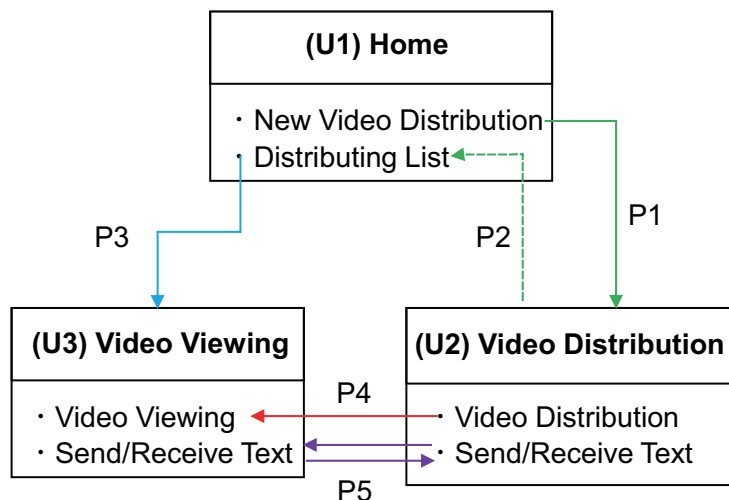

**Figure 7.** Overview of Web screen transitions in the proposed method.

**Step 8: Introduction of WebSocket Technology**

In Step 8, WebSocket technology is introduced in the P2 and P5 sections described in Step 7. Specifically, for P2, the ID of the URL corresponding to the new transmission is prepared on the U2 client-side, and the client-side of U1 receives the ID and assigns it to the URL for viewing via a WebSocket server. On the other hand, for P5, set the methods to send and receive new messages on the client-side of U2. Meanwhile, it sends them to the client-side of U1 via the WebSocket server. By going through the WebSocket server, the

delivery content with the specified ID and the content of the receiving and sending can be shared on the viewing page.

**Step 9: Introduction of PeerJS technology**

In Step 9, PeerJS technology is introduced in the P4 section described in Step 7. In conventional PeerJS technology, bidirectional communication is expected. The proposed system requires unidirectional communication to distribute the video stream from the hospital to the classroom. Hence, the communication process of PeerJS technology must be devised. Our key idea is to apply image processing technology to the hospital-side video stream, convert it into a canvas stream, and distribute it to the classroom side. However, we do not distribute the video stream from the classroom side to the hospital side but instead create an empty video stream by ourselves, thereby eliminating this process. This way makes P4 a unidirectional communication, and communication from the classroom side to the hospital side can be achieved only by receiving and sending texts using WebSocket.

*3.5. System Implementation*

We use the following hardware to achieve a secure streaming system: (1) General-purpose computer: Lenovo IdeaPad C340. (2) USB fixed-point camera: Logitech OEM B500. In order to validate the effectiveness of the proposed method, we implement a secure streaming system using the following techniques (see Table 2). An example of the homepage of the proposed system is shown in Figure 8a, and the video streaming page and viewing scene are shown in Figure 8b,c, respectively (refer to Figure 7). We have added checkboxes for "eye mosaic" and "face mosaic" and a new button for the video delivery on the homepage so that users can choose according to their video delivery needs. In addition, the name of the subject, time of delivery, and the number of viewers of the delivered video can be automatically updated in the list of available videos on the homepage by recording and statistics by the WebSocket server. In addition, the video viewing page requires the user to manually press a button to disable audio generation due to a Web browser functionality limitation in the case of automatic media generation.

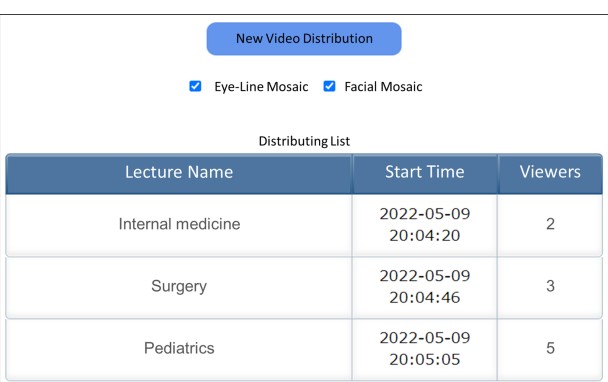

(a)

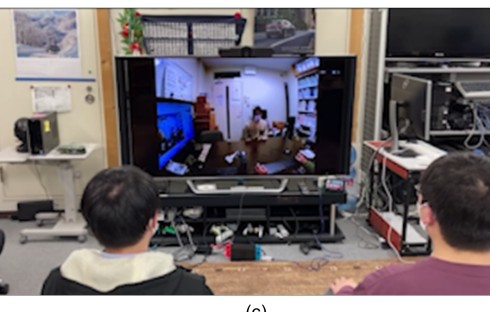

(c)

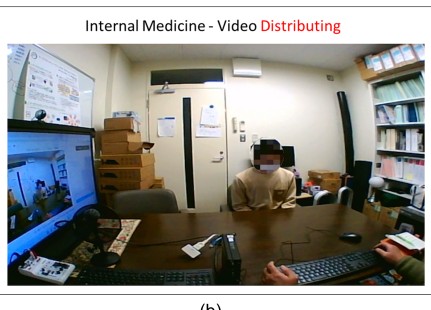

(b)

**Figure 8.** Web UI and scene: (**a**) Home page. (**b**) Video distribution page. (**c**) Video viewing scene.

**Table 2.** Technologies used in the implementation of the proposed system.

| Item | Technologies and Libraries Used |
|---|---|
| HTTP server | `express`, `http` |
| WebSocket technology | `socket.io` |
| PeerJS technology | `peer` |
| Random ID generation | `uuid` |
| Basic authentication | `basic-auth-connect` |
| Bone sensing | PoseNet Model |
| Web browser | Chrome (101.0.1951.54) |

## 4. Preliminary Experiment

This section introduces results, discussion, and a preliminary experiment conducted for comparing the face masking performance between bone sensing and face recognition.

### 4.1. Purpose and Experimental Setup

The purpose of this section is to compare the effectiveness of the methods proposed in Section 3 from various perspectives. Specific experimental settings are shown in Table 3. We use the proposed method to condition different viewing angles and interpersonal distances of human faces photographed by a depth camera (see Figures 9 and 10). Based on this, we are trying to figure out to what size the proposed method can mask the face for each successive image. Moreover, we do not evaluate only the proposed method but also collect the results of experiments with the same approach as the proposed method using features output from recent face recognition technology (see Figure 11). We compare and discuss bone sensing and face recognition results. In this experiment, all subjects provided consent to the processes. Our goal is to record 140 frames (i.e., roughly 5 s) in each of five different head directions (i.e., front, top, down, left, right) at three different distances (i.e., near, middle, far) from the person to the camera. We used a depth camera to record integrated movies (i.e., $140 \times 5 = 700$ frames) of five different head orientations at three different subject positions at three different distances and the distance data measured for each. The video recording and distance data measurement were realized using Python programming on Jupyter Notebook [22] with a general-purpose computer and introducing two libraries: OpenCV [23] and Pyrealsense2 [24].

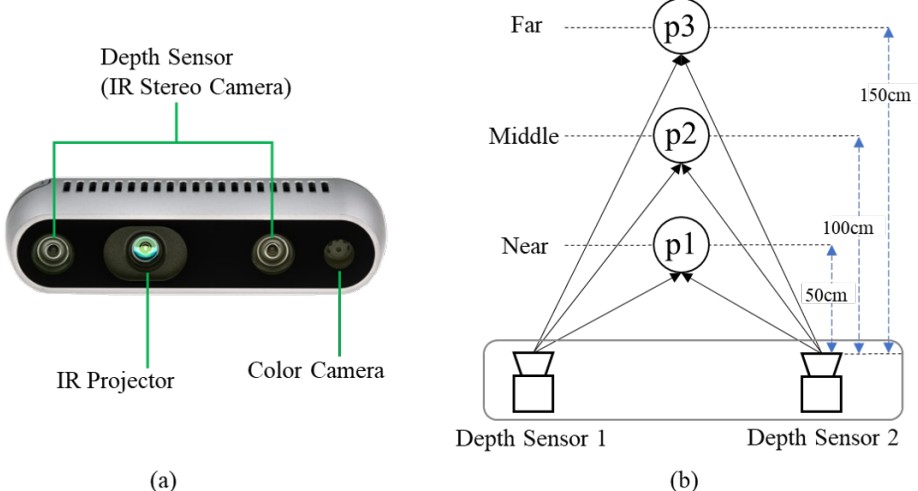

(a)  (b)

**Figure 9.** Experimental tools and settings: (**a**) Intel RealSense Depth Camera D435. (**b**) Experimental settings with three distances.

**Table 3.** Experimental settings.

| Items | Contents |
| --- | --- |
| Target space | One seat in ES4 Nakamura lab |
| Subject information | One person (male, 20 s) |
| Video total length | 700 frame |
| Video resolution | 640 × 480 |
| Frame time interval | 30 fps |
| Shooting method | Intel RealSense Depth Camera D435 [25] |
| Number of head directions | 5 |
| Head directions | Front, up, down, left, right |
| Video length of each head direction | 140 frames |
| Number of distances from person to camera | 3 |
| Distances from person to camera | About 50 cm (near), 100 cm (middle), 150 cm (far) |
| Used technologies | Bone sensing (posenet.js), face recognition (face-api.js) |
| Number of masking face methods | 2 |
| Masking face methods | Eye line mosaic, face mosaic |

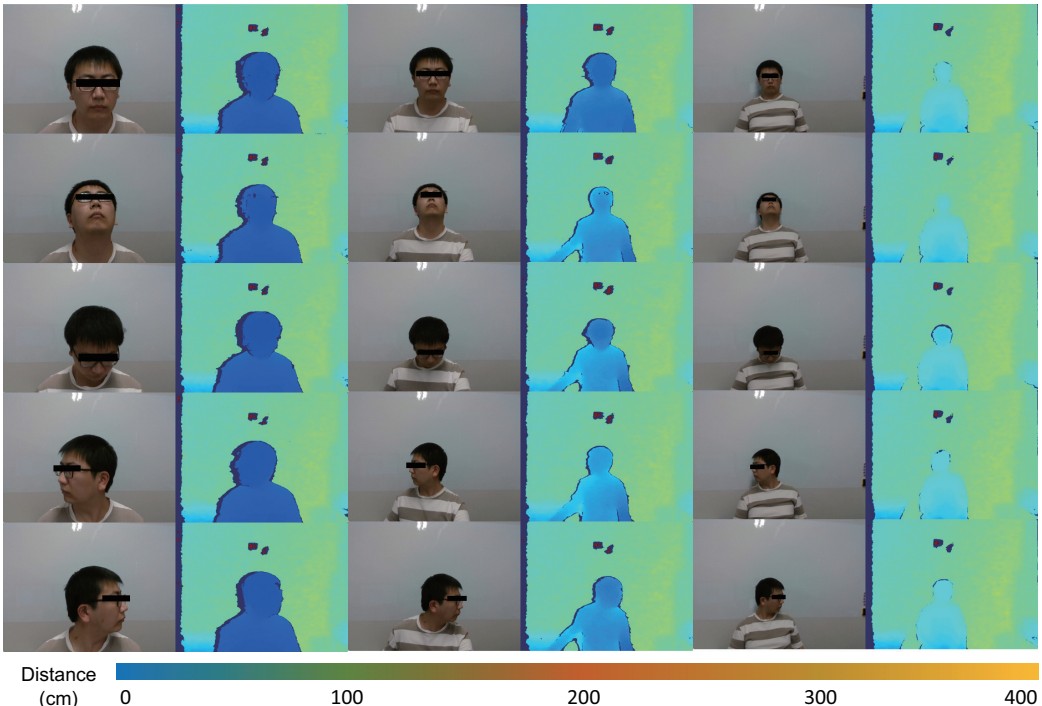

**Figure 10.** Representative eye line mosaic effects using bone sensing and depth camera.

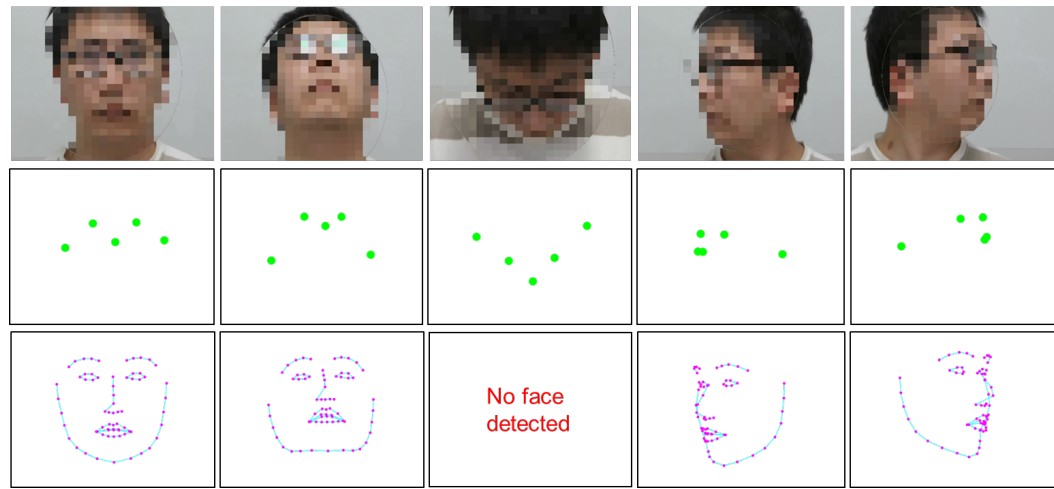

**Figure 11.** Outputting feature positions from bone sensing and face recognition, respectively.

### 4.2. Evaluation Method and Scale

We visualize the size of facial masking and evaluate it from two aspects: continuity and stability. For each of the 700 frames of video at each distance described in Section 4.1, we perform two types of approaches using bone sensing: an eye mosaic and a face mosaic. To effectively evaluate consecutive facial masking, we record the size (i.e., length and width) of the mosaic area output from each uproot and merge it with the distance data in a time series. Hence, we can expect to visualize the size change of the eye and face mosaics on the frame for each of the five different head orientations of the subject (i.e., front, top, up, left, and right, 140 frames each) at the same distance. In this experiment, we set the width of the eye mosaic to 0.2 times the length and the size granularity of the face mosaic to 20 pixels. Meanwhile, we introduce a model of face recognition, face-api.js [26], with an approach similar to the proposed method (see Figure 12). This model can be run locally as a trained model, similar to bone sensing, and can estimate and output 68 coordinate values at human facial landmarks from images and video streams. For each of the 700 frames of video at each distance described in Section 4.1, we also perform two approaches to face recognition: an eye mosaic and a face mosaic. Hence, we compare our proposed method of facial masking by bone sensing with facial masking by face recognition and provide a quantitative evaluation of consecutive facial masking.

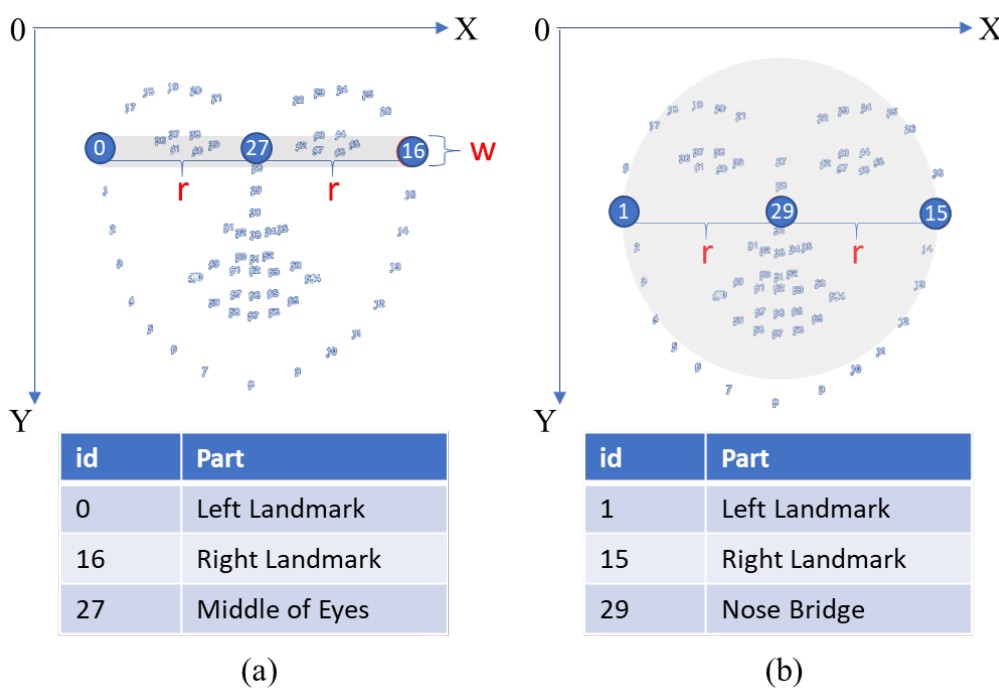

**Figure 12.** Approaches using face recognition: (**a**) Eye line mosaic. (**b**) Face mosaic.

### 4.3. Results Using Image-Based Bone Sensing

Figure 13 shows the results of facial masking using image-based bone sensing in this experiment. The three graphs on the left side of Figure 13 show the mosaic size results for each distance in the eye line mosaic, and the three on the right side show the mosaic size results for each distance in the face mosaic. When the distance is near, and the head direction is front, the length and width of eye line and face mosaics are around 10. When the head direction is up and left at a near distance, the length and width of the eye line mosaic are reduced by about 40 and 10, respectively, and the length and width of the face mosaic are reduced by about 20 to 30. When the head direction is down and to the right at a near distance, the eye line mosaic length and width are increased by about 30 and 10, respectively, and the face mosaic length and width are increased by about 20 to 50. On the other hand, the length and width of eye line and face mosaics are changed by about 10 to

20 at the middle distance and when the head direction is front, top, and down. Meanwhile, the length and width of the eye line mosaic are reduced by about 5 to 20, and the length and width of the face mosaic are intensified by about 20 to 50 when the head direction is left or right at the middle distance. Furthermore, when the distance is far, and the head direction is front and up, the length and width of the eye line mosaic are reduced, and the length and width of the face mosaic are changed around 10. When the head is down at a far distance, the length and width of both the eye line and face mosaics are reduced by about 20. When the head direction is left, and right at the far distance, the length and width of both eye line and face mosaics are increased by about 10 to 20.

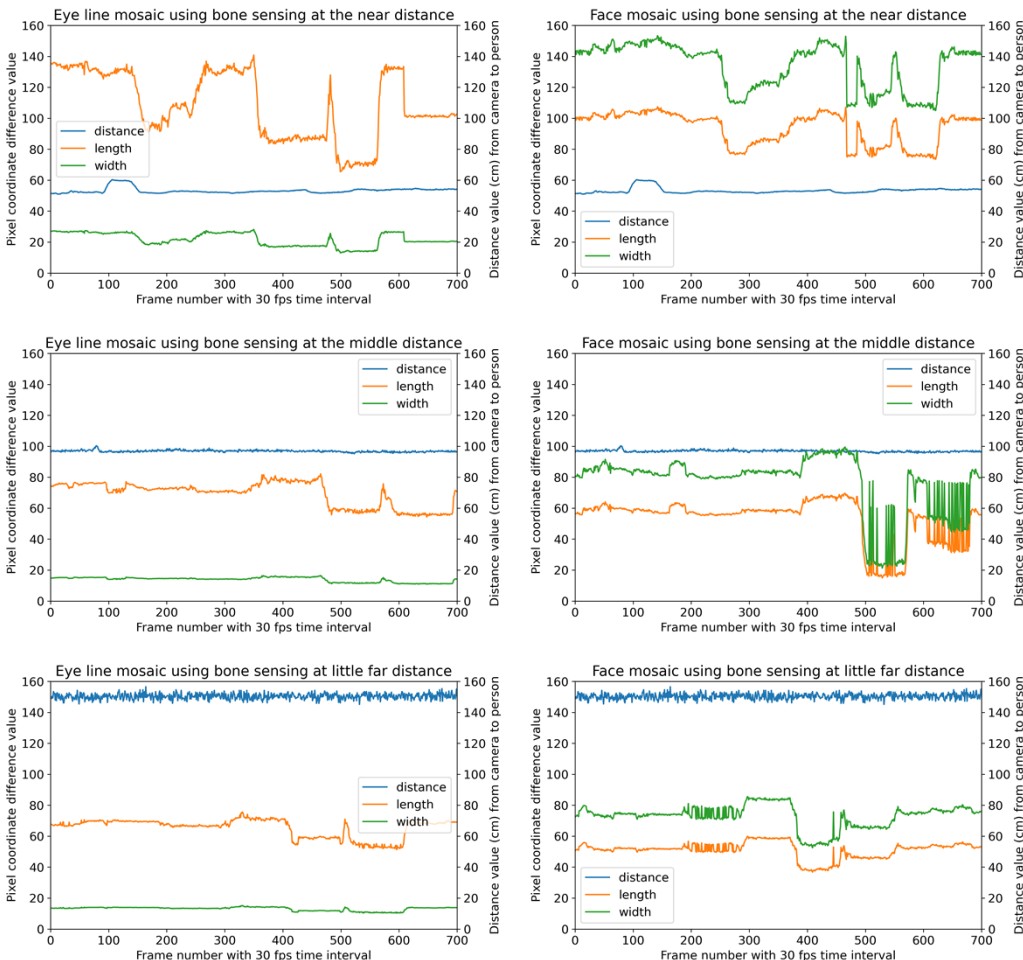

**Figure 13.** Facial masking results using image-based bone sensing in this experiment.

### 4.4. Results Using Image-Based Face Recognition

As a contrasting result, Figure 14 shows the results of facial masking using image-based face recognition in this experiment. Similar to the distribution of results in Figure 13, the three graphs on the left side of Figure 14 show the results for each distance mosaic size in the eye line mosaic, and the three graphs on the right side show the results for each distance mosaic size in the face mosaic. When the head is in the frontal direction at a near distance, the length and width of both the eye line and face mosaics show a variation of around 10. When the head direction is up and to the right at a near distance, the length and width of the eye line mosaic are reduced to 0. The length and width of the eye line mosaic are increased by about ten, and the length and width of the face mosaic are increased by about 5 to 30 when the head direction is down and left at a near distance. On the other hand, the length and width of the eye line mosaic and face mosaic are not changed much when the head direction is front and up at the middle distance. Meanwhile, both the eye

line mosaic and the face mosaic are reduced to 0 when the middle distance and the head direction are down. Moreover, when the head direction is left, and right at the middle distance, the length and width of the eye line mosaic are increased by about 20 to 30, and the length and width of the face mosaic are changed by about 10. When the head direction is up, left, and right at a far distance, the length and width of both eye line and face mosaics show about 10 to 20 variations. When the head is down at a far distance, the length and width of both eye line and face mosaics are intensified by about 40. When the distance is far away, and the head direction is down, the length and width of the eye line mosaic will be reduced to 0.

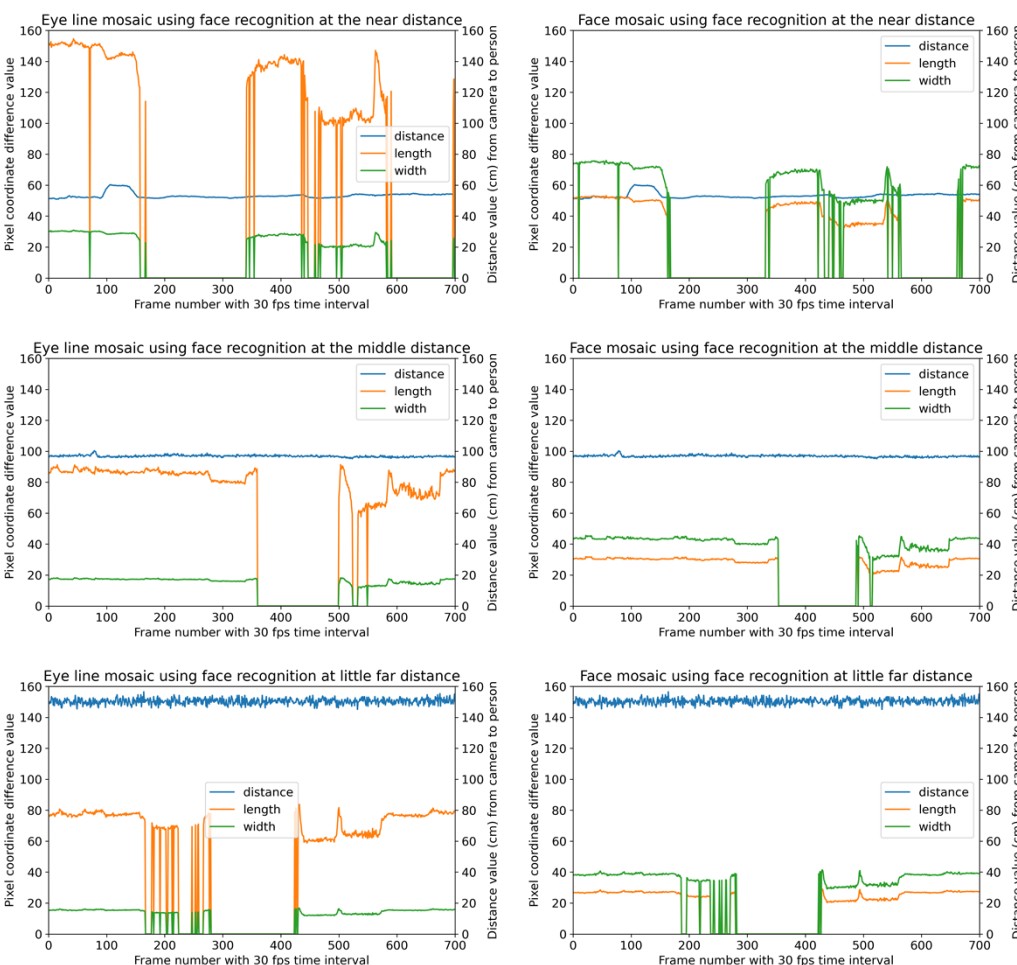

**Figure 14.** Facial masking results using image-based face recognition in this experiment.

### 4.5. Experimental Discussion

To further elucidate the results of this experiment, we combined six parameters: head direction, distance, eye line mosaic, face mosaic, bone sensing, and face recognition. Figure 15 shows each case's box-and-whisker plot of the resulting mosaic pixel area. The upper two graphs in Figure 15 show the results of eye line and face mosaicing using bone sensing, and the lower two graphs show face recognition results. The bone sensing results show that at near distances, the eye line mosaic changes most intensely when the head direction is down, and the face mosaic changes most intensely when the head direction is left and right. We also found that the eye line mosaic was less intense and more stable at middle and far distances. We also observed changes in the face mosaic at middle distances with head direction left and right and far distances with head direction down. On the other hand, the results using face recognition show that the eye line mosaic and face mosaic are stabilized at near distances and when the head direction is front. However, otherwise, the mosaic can mostly not have a more consecutive effect. In particular, eye line and face

mosaics were often severely altered or not mosaiced at middle and far distances and when the head direction was down, left, or right. Our experiments found that bone sensing is more effective than face recognition for consecutive facial masking by combining various factors such as distance and head direction; however, we assume that the complexity of the facial masking process is further increased when there is more than one person. We believe that it is highly likely that the distance from the camera to each subject is different, and a combined approach of eye line and face mosaicing may be better.

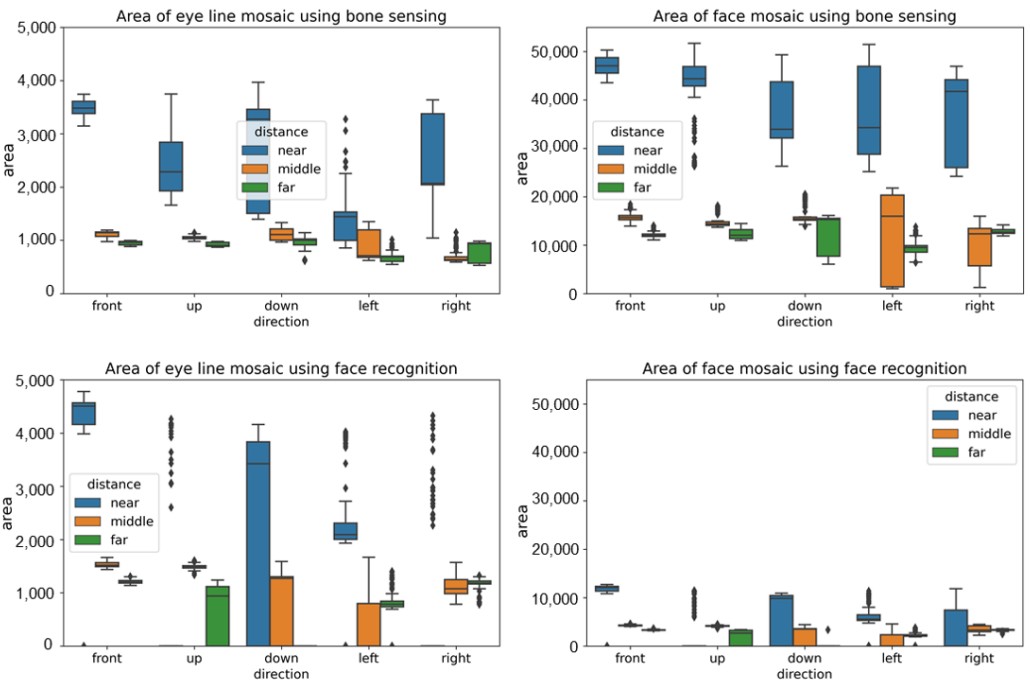

**Figure 15.** Comparing area of masking face with different directions and distances.

## 5. Conclusions

In this paper, we developed a secure streaming system using image-based bone sensing to evaluate consecutive and effective face masking. In our experiment, for consecutive face masking, the results using bone sensing are unaffected by distance and head orientation; their variation width of the mosaic area is stable within around 30% of the target area; however, about three-fourths of the results using conventional face recognition were unable to mask their faces consecutively. The main contributions of this study are as follows: (1) A secure streaming system was developed and implemented by integrating image-based bone sensing and WebRTC technologies. (2) We quantitatively evaluated the size variation of face masking using bone sensing by adding factors such as distance, head direction, and masking method. (3) A similar approach using general face recognition technology was also implemented and compared with the results using bone sensing to verify the effectiveness of our proposal; however, there are some technical limitations in this study. For example, it is necessary to evaluate other related factors such as brightness around the camera, recognition accuracy, and multiple users. In addition to supporting remote medicine education, we also envision the challenge of real-time face masking during live surveillance and interviews. In future work, we organize the elements and objectives described above and design algorithms to perform lightweight face masking processing in more complex cases [27]. We also envision that the algorithms can be used to mask not only human faces but also user-defined objects in real time, which can be useful in protecting the privacy of valuable items. In addition to privacy protection, we would like to devise a method for communication security and consider the possibility of secure video distribution without using an internal network.

**Author Contributions:** Writing—original draft preparation, S.C.; writing—review and editing, S.C.; supervision, M.N. and K.S.; validation, S.C. All authors have read and agreed to the published version of the manuscript.

**Funding:** This research received no external funding.

**Institutional Review Board Statement:** Not applicable.

**Informed Consent Statement:** Not applicable.

**Acknowledgments:** This research was partially supported by JSPS KAKENHI Grant Numbers JP19H01138, JP20H05706, JP20H04014, JP20K11059, JP22H03699, JP19K02973, Grant-in-Aid for JSPS Research Fellow (No.22J13217), and Tateishi Science and Technology Foundation (C) (No.2207004).

**Conflicts of Interest:** The authors declare no conflict of interest.

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
