# Peer review of "Consecutive and Effective Facial Masking Using Image-Based Bone Sensing for Remote Medicine Education"

_applsci, doi:10.3390/app122010507_

Round 1

Reviewer 1 Report

The work is very good.

I suggest some changes in the structure of the work.

1. Item 2 must be part of the introduction.

2. Between items 3 and 3.1 there should be a small induction anticipating what will come in the item.

3. Likewise between 4 and 4.1.

4. I believe that item 4.5 can be placed in the previous items, if the authors find it useful.

5. The conclusion should have some quantitative data, if possible.

6. The figures are too small, I suggest trying to increase the size.

Reviewer 2 Report

The manuscript presents a solution to face/eye mosaic for the online streaming of medical education. Whilst the research problem itself is interesting, the work described in the manuscript is mostly about engineering implementation. As a research paper, it is supposed to focus more on the research problem and how to address the problem. In this work, the research problem could be, for example, the detection of faces/eyes. Even though using an existing technique such as PoseNet is acceptable, the authors should have conducted more extensive experiments to validate the performance of the algorithm or the whole system. I don't see experiments with valid settings and any quantitative results to validate the effectiveness/robustness/accuracy of the proposed solution in the current manuscript. For the qualitative evaluation, one should also demonstrate some failure cases and analyse why they fail.

There are also some language issues, e.g., ln 64: reminder -> remainder

Reviewer 3 Report

Following comments to be strictly adressed:

1.      Abstract need to be revised with quantitative results.

2.      How your work analyzes the factors like camera-to-human distance, head direction, and mosaic schemes and provided optimal values for readers need to be reported with supporting literatures.

3.      What are the other factors which present technical challenges need to be discussed, some where in introduction? Proper justification needed to be given why you neglected such factors in your analysis.

4.      Does your research analyse the all-age group people.

5.      The camera specification needed to be described clearly in the appendix.

6.      Group citations to be strictly avoided, and highlight the individual contributions. For example: [3-5], [8-11], [12-16], [31-35].

7.      In Fig. 4 Which machine learning algorithm is used need to be clearly defined.  Please also describe with scientific justification of selection of the particular algorithm for your research work.

Reviewer 4 Report

The author proposed their Consecutive and Effective Facial Masking Using Image-Based Bone Sensing for Remote Medicine Education. I have following comments before proceedings further process:

1. I think this paper is belongings to application oriented paper. I am not sure about this paper which suite for this journals.

2. No new novelty in this paper. Only face masking is done. 

3. The author didnt compared exisiting works. 

4. There will be lack uses of new technologies and new innovations.

Round 2

Reviewer 4 Report

I have no comments 
